# Medicinal Herbs in the Relief of Neurological, Cardiovascular, and Respiratory Symptoms after COVID-19 Infection A Literature Review

**DOI:** 10.3390/cells11121897

**Published:** 2022-06-11

**Authors:** Joanna Nawrot, Justyna Gornowicz-Porowska, Jaromir Budzianowski, Gerard Nowak, Grzegorz Schroeder, Joanna Kurczewska

**Affiliations:** 1Department and Division of Practical Cosmetology and Skin Diseases Prophylaxis, Poznan University of Medical Sciences, Rokietnicka 3, 60-806 Poznan, Poland; justyna.gornowicz-porowska@ump.edu.pl (J.G.-P.); jbudzian@ump.edu.pl (J.B.); gnowak.gerard@gmail.com (G.N.); 2Faculty of Chemistry, Adam Mickiewicz University in Poznan, Uniwersytetu 5, Poznanskiego 8, 61-614 Poznan, Poland; schroede@amu.edu.pl (G.S.); asiaw@amu.edu.pl (J.K.)

**Keywords:** COVID-19, complications, viral infections, phytotherapy, medicinal herbs

## Abstract

COVID-19 infection causes complications, even in people who have had a mild course of the disease. The most dangerous seem to be neurological ailments: anxiety, depression, mixed anxiety–depressive (MAD) syndromes, and irreversible dementia. These conditions can negatively affect the respiratory system, circulatory system, and heart functioning. We believe that phytotherapy can be helpful in all of these conditions. Clinical trials confirm this possibility. The work presents plant materials (*Valeriana officinalis*, *Melissa officinalis*, *Passiflora incarnata*, *Piper methysticum*, *Humulus lupulus*, *Ballota nigra*, *Hypericum perforatum*, *Rhodiola rosea*, *Lavandula officinalis*, *Paullinia cupana*, *Ginkgo biloba*, *Murraya koenigii*, *Crataegus monogyna* and *oxyacantha*, *Hedera helix*, *Polygala senega*, *Pelargonium sidoides*, *Lichen islandicus*, *Plantago lanceolata*) and their dominant compounds (valeranon, valtrate, apigenin, citronellal, isovitexin, isoorientin, methysticin, humulone, farnesene, acteoside, hypericin, hyperforin, biapigenin, rosavidin, salidroside, linalool acetate, linalool, caffeine, ginkgolide, bilobalide, mihanimbine, epicatechin, hederacoside C,α-hederine, presegenin, umckalin, 6,7,8-trixydroxybenzopyranone disulfate, fumaroprotocetric acid, protolichesteric acid, aucubin, acteoside) responsible for their activity. It also shows the possibility of reducing post-COVID-19 neurological, respiratory, and cardiovascular complications, which can affect the functioning of the nervous system.

## 1. Introduction

The coronavirus disease 2019 (COVID-19) caused by the severe acute respiratory syndrome coronavirus 2 (SARS-CoV-2) has rapidly spread worldwide. COVID-19 clinical presentation primarily affects the respiratory system, but it is increasingly being recognized as a systemic disease with neurologic manifestations. Research shows that the many neurological symptoms of COVID-19 are likely a result of the body’s overall immune response to infection rather than the virus directly infecting the brain or nervous system [1]. Based on previous coronavirus infection outbreaks, more than one-third of SARS and MERS survivors suffered from neuropsychiatric problems, such as stress, anger, and depression for >6 months and impaired quality of life for ≥12 months. That is why it is worth focusing on treating long-term, lasting neurological complications and other health problems. Controlling the inflammatory reaction may be as important as targeting the virus. [1,2].

Medicinal plants are widely used to treat a variety of infectious ailments. 25% of commonly used drugs contain compounds isolated from plants [3]. A variety of medicinal plants have shown promise in the treatment of many viral infections. The development of vector-based strategies and advances in the separation of active compounds from biomass gives the possibility of the discovery of modern drugs. Mukhtar et al. [3] describe the potential antiviral properties of medicinal plants against a diverse group of viruses and suggests screening the potential of plants with a broad spectrum of antiviral activity against emerging viral infections.

The severity of the COVID-19 infection ranges from mild symptoms such as fever, cough, and dizziness to severe illness and even death. The oldest medical pharmacopeias of the African, Arabian, and Asian countries solely utilize plants and herbs to treat pain, oral diseases, skin diseases, microbial infections, multiple types of cancers, and reproductive disorders, among a myriad of other ailments. The World Health Organization (WHO) estimates that over 65% of the world population solely utilizes botanical preparations as medicine [4].

The phytochemical molecules of antiviral activity extracted from medicinal plants have been reported as possible therapeutic agents in treating COVID-19 disease [5].

Sadeek and Abdallah [5] believe, on the basis of the knowledge of other epidemics, that herbal medicines have been long used to attenuate infectious diseases because of their lower side effects, low cost, and reduced potential to cause resistance. Observations from traditional medicine revealed that many plant-derived essential oils could effectively prevent and treat viral-induced respiratory tract infections. In a review paper, Salem and Ezzat [6] presented the screening use of essential oils to prevent or treat COVID-19.

Studies have confirmed the role of many plants against respiratory viruses when employed either as crude extracts or their active ingredients in pure form. A review article by Khan et al. [7] highlights the importance of phytomedicine against COVID-19 and presents a review of the mechanistic aspects of the essential phytochemical compounds that have shown potential against coronaviruses. Glycyrrhizin from the roots of *Glycyrrhiza glabra* has demonstrated promising potential against the previously epidemic coronavirus, SARS-CoV. Other important plants such as *Artemisia annua*, *Isatis indigotica*, *Lindera aggregate*, *Pelargonium sidoides*, and *Glycyrrhiza* spp. have been employed against SARS-CoV. Active ingredients (emodin, reserpine, aescin, myricetin, scutellarin, apigenin, luteolin, and betulonic acid) have shown promising results against the coronaviruses. Phytochemicals have demonstrated activity against the coronaviruses through mechanisms such as viral entry inhibition, inhibition of replication enzymes, and virus release blockage.

Bachar et al. [8] present a review attempt that has been taken to summarize the medicinal plants reported for exhibiting antiviral activities available in Bangladesh, along with discussing the mechanistic insights into their bioactive components against the three most hazardous viruses, namely SARS-CoV-2, HIV (Human Immunodeficiency Virus), and HBV (Hepatitis B virus). The review covers 46 medicinal plants with antiviral activity from 25 families. Hesperidin, apigenin, luteolin, seselin, 6-gingerol, humulene epoxide, quercetin, kaempferol, curcumin, and epigallocatechin-3-gallate (EGCG) have been reported to inhibit multiple molecular targets of SARS-CoV-2 viral replication in a number of in silico investigations. In addition, numerous in silico, in vitro, and in vivo bioassays have demonstrated that EGCG, anolignan-A and B, ajoene, curcumin, and oleanolic acid exhibit anti-HIV activity, while piperine, ursolic acid, oleanolic acid, (+)-cycloolivil-4′-*O*-β-d-glucopyranoside, quercetin, EGCG, kaempferol, aloin, apigenin, rosmarinic acid, andrographolide, and hesperidin possess anti-HBV activity. Thus, the antiviral medicinal plants and the isolated bioactive compounds may be considered for further advancing the investigations to develop effective and affordable antiviral drugs.

Plant-based decoctions have been a significant part of Jamaican traditional folklore medicine. Jamaica is of particular interest because it has approximately 52% of the established medicinal plants that exist on earth. Lowe et al. [4] offer a review of some important Jamaican medicinal plants, with a particular reference to their antiviral activity.

Traditional Chinese medicine (TCM) showed appreciable results in improving clinical symptoms and reducing mortality and recurrence rates of the virus [9]. A traditional Chinese medicine formula, Taiwan ChingguanYihau (NRICM101), has been administered to patients with COVID-19 in Taiwan since April 2020. NRICM101 is based on a 470-year-old TCM formulation known as jingfangbaidusan, which NRICM adapted to address the variability of COVID-19 and its differences from SARS, as well as the reparation’s clinical performance. Pharmacological assays demonstrated the effects of the formula in inhibiting the spike protein/ACE2 interaction, 3CL protease activity, viral plaque formation, and production of cytokines interleukin (IL)-6 and tumor necrosis factor (TNF)-α. This bedside-to-bench study suggests that NRICM101 may disrupt disease progression through its antiviral and anti-inflammatory properties, offering promise as a multitarget agent for the prevention and treatment of COVID-19 [10].

### COVID-19 Related Complications in Humans

COVID-19 primarily affects the respiratory system, causing severe pneumonia and acute respiratory distress syndrome in severe cases; it can also result in multiple extrapulmonary complications. The pathogenesis of extrapulmonary damage in patients with COVID-19 is probably multifactorial, involving both the direct effects of SARS-CoV-2 and the indirect mechanisms associated with the host inflammatory response [11]. The majority of the patients with COVID-19 have fever and cough. Critically ill patients can develop dyspnea and acute respiratory distress syndrome [12].

Although SARS-CoV-2 mainly causes respiratory diseases, growing data indicate that SARS-CoV-2 can also invade the central nervous system (CNS) and peripheral nervous system (PNS), causing multiple neurological diseases, such as encephalitis, encephalopathy, Guillain–Barré syndrome, meningitis, and skeletal, muscular symptoms [13]. Coronaviruses were reported to reach the CNS, causing neurovirulence [14]. However, the exact mechanism by which coronaviruses invade the CNS has not been entirely clear [15]. Mostly, viral infections start from peripheral tissues and then spread to the peripheral nerves and finally reach the central nervous system [16]. This process may explain the presence of neurological lesions such as demyelination [17]. ACE2 is highly expressed on vascular endothelial cells and also expressed on olfactory epitheliums, striatum, cortex, substantia nigra, as well as the brainstem, suggesting SARS-CoV-2 can directly infect vascular endothelial cells to cross the blood–brain barrier and then can infect cells throughout the CNS [18].

There are reports of nervous disorders caused by ongoing or past COVID-19 infection. During hospitalization, many patients experienced neurological symptoms; the most common were fatigue, decreased mood, and anxiety. Non-hospitalized COVID-19 “long haulers” experience prominent and persistent “brain fog”, fear, anxiety, and fatigue that affect their cognition and quality of life [19].

Elderly individuals are especially at high risk of developing severe forms of COVID-19 because of the factors associated with aging and a higher prevalence of medical comorbidities. Therefore, they are more vulnerable to possible lasting neuropsychiatric and cognitive impairments. Several reports have described insomnia, depressed mood, anxiety, post-traumatic stress disorder, and cognitive impairment in a portion of patients after discharge from the hospital. The potential mechanisms are probably multifactorial, involving the direct neurotrophic effect of SARS-CoV-2, consequences of long intensive care unit stay, mechanical ventilation, brain hypoxia, systemic inflammation, and secondary effects of medications used to treat COVID-19. Chronic diseases such as dementia are a particular concern not only because they are associated with higher rates of hospitalization and mortality but also because COVID-19 further exacerbates the vulnerability of those with cognitive impairment. In patients with dementia, COVID-19 frequently has an atypical presentation with mental status changes, complicating cases’ early identification [20]. Therefore, it seems justified to use safe and effective herbal medicines as prophylactic protection against the risk of dementia.

The anxiety states are accompanied by dizziness, shortness of breath, rapid breathing, and easy fatigue. When prolonged, such states lead to a drop in mood, sadness, loss of joy, fatigue, and a lack of willingness to act. These are significant symptoms of the onset of depression. Faster diagnosing of the symptoms is very important. Inadequate administration of sedatives will not only aggravate the symptoms of depression but also become dangerous due to inhibition of the functioning of the central nervous system as a result of prolonged intensive care. Anxiety disorders are associated with deregulation of neurotransmission in the central nervous system and disruption of neurobiological processes responsible for controlling stress responses [21].

The new diagnostic unit is mixed anxiety-depressive disorders (MAD). Symptoms such as sleep disorders, mood swings, and anticipating the worst are common to anxiety and depression [22]. MAD is often accompanied by difficulty in falling asleep. It seems reasonable to suspect that changes in insomnia could predict changes in MAD prospectively. Targeting insomnia in the context of brief cognitive behavioral therapy, with the help of herbal remedies that facilitate falling asleep and have an antianxiety effect, in persons with mild to moderate anxiety and depression, may further reduce not only symptoms of insomnia but also symptoms of anxiety and depression [23].

## 2. Phytotherapy of the Central Nervous System

In the physiological picture of depression, anxiety disorders, and stress, there is an increased secretion of corticoliberin. This is a result of stimulation of the hypothalamus. Corticoliberin releases more significant amounts of ACTH (corticotropin), which acts on the adrenal cortex and causes cortisol secretion. Under physiological conditions, its higher amount causes inhibition of hypothalamic–pituitary–adrenal axis excitation. However, under prolonged stress and depression, this reaction stops functioning—hypercortisolemia occurs, and, as a consequence, hippocampal neurons may be damaged, causing memory and concentration disorders. These are typical symptoms of chronic stress, anxiety, and depression [24].

COVID-19 destabilized global healthcare and forced medical professionals to provide treatment in previously uncommon manners. Berlinska et al. [25] proposed a new diagnostic algorithm for hypercortisolemia in the COVID-19 times. It became essential to identify cases requiring urgent medical attention and to offer watchful waiting to mild or doubtful cases. Medical treatment of hypercortisolemia and secondary comorbidities became especially important.

Plant-based drugs acting on the central nervous system are an important alternative to synthetic drugs. The patient’s age and concomitant diseases require a well-tolerated treatment and minimal adverse effects on the body. In addition, they are at least as effective as synthetic drugs.

### 2.1. Plant Raw Materials with an Anxiolytic Effect

Gamma-aminobutyric acid (GABA, Figure 1) is a neuromediator in the central nervous system responsible for the inhibition of anxiety. This compound plays an important role in the process of reducing overexcitation [26]. Gamma-aminobutyric acid is a naturally occurring amino acid that works as a neurotransmitter in the brain. GABA is considered an inhibitory neurotransmitter because it blocks or inhibits certain brain signals and decreases activity in the nervous system. When GABA attaches to a protein in the brain known as a GABA receptor, it produces a calming effect. This can help with feelings of anxiety, stress, and fear. It may also help to prevent seizures. Before or after the onset of viral inflammation symptoms, oral GABA supplements effectively reduced MHV-1 pneumonia, severe disease, and death. GABA very effectively limited MHV-1-inducing pneumonitis, severe illness, and death. GABA treatment also reduced viral load in the lungs, suggesting that GABA-Rs may provide a new druggable target to limit coronavirus replication [27].

Bruni et al. [28] presented a systematic analysis of studies assessing the mechanisms of action of various herbal medicines on different subtypes of GABA receptors in the context of sleep control. Currently, available evidence suggests that herbal extracts may exert some hypnotic and anxiolytic activity by interacting with GABA receptors and modulating GABAergic signaling in the brain. Still, their mechanism of action in the treatment of insomnia is not entirely understood.

Herbal anxiolytics may be the first choice for early anxiety states. Unlike benzodiazepines, plant ingredients with anxiolytic and sleep-promoting properties do not cause addiction, tolerance, withdrawal, insomnia rebound effect, and respiratory depression.


***Valerianae radix*-*Valeriana officinalis* L.**


Characteristic compounds (Table 1)

Sesquiterpenes–valeranone

Irydoids–valtrate

In in vitro studies, hydroalcoholic and aqueous extracts of valerian root interacted with the GABA - benzodiazepine chlorine channel receptor complex in isolated brain cells. They inhibited reuptake and induced the release of [^3^H]-labeled GABA in synaptosomes (axon terminal sites). The same extracts displaced labeled muscimol (an alkaloid-a selective agonist of GABA_A_ receptor). These reactions were Na^+^ cation dependent and independent of Ca^++^ cations in the extracellular fluid. This indicates that valerian root extracts release GABA by reversing its transport. Research has also shown that alcohol extracts (free from γ-aminobutyric acid) exhibit similar properties. On the other hand, biochemical studies of valerian root extracts showed their affinity for the following receptors: adenosine A1 (the excitability of dendrites in neurons is inhibited, resulting in anticonvulsant and sedative effects), 5-HT_1A_ (agonists of this subreceptor, present in the hippocampus, have an anxiolytic and antidepressant effect), and two melatonin receptor subtypes (ML_1_ and ML_2_) [29].

Valeranone administered parenterally in in vivo studies showed the effects on the CNS as a depressant and smooth muscle relaxant. Valeranone also decreased the spontaneous mobility of the test animals 5 h after administration. Valerian root tincture showed similar properties. Recent studies of valerian root extract showed no sedative, hypnotic, or relaxant effects, while the authors of this study concluded that it had significant anxiolytic and antidepressant activity [30].

In human pharmacological studies, valerian root extract (1.2 g single dose) compared with 10 mg of diazepam found the effectiveness of a herbal drug in comparison with a synthetic drug for ⅓. Another study showed a statistically significant reduction in systolic blood pressure and heart rate after using 2 × 300 mg daily of valerian dry ethanolic extract for 7 days in volunteers undergoing mental stress [31].

There was no daytime sleepiness after using valerian root extract. This has been confirmed in several human pharmacological trials. Based on the bibliographic data, it can be assumed that valerian root preparations are an effective anxiolytic drug, devoid of side effects such as drowsiness, poor concentration, and memory disorders. Therefore, they can be used for the first symptoms of anxiety disorders [32].

After 14 days of administration of 600 mg of dry extract, no delayed reaction time, memory impairment, or loss of alertness were observed. However, the latter symptom was observed 1–2 h after the administration of the syrup of the material in question. Therefore, driving vehicles and dangerous machinery is not recommended within two hours of applying the preparation with valerian root [33].

There are no data on the use of the drug during pregnancy, breastfeeding, and in children under 3 years of age [34].


***Melissae folium-Melissa officinalis* L.**


Characteristic compounds (Table 1)

Flavonoids–apigenin

The flavonoids are probably responsible for the anxiolytic effect of lemon balm leaves. In in vivo tests, lemon balm leaves reduced the feeling of danger and anxiety, facilitated falling asleep, and counteracted digestive tract disorders caused by nervous tension [35]. Lemon balm leaf probably caused a depressant effect of mint leaves on the CNS in preparations composed of both raw materials [36]. In human pharmacological studies, single-dose administration of lemon balm leaf extract may modulate mood and cognition in healthy young volunteers. It has also been shown that lemon balm leaf extracts can reduce negative mood changes associated with 20 min of induced psychological stress [37]. The use in children under 12 years of age has not been established due to a lack of adequate data. There are no data on the possibility of use during pregnancy and lactation. There are no restrictions on the length of use of the raw material [38].

***Passiflorae herba–Passiflora incarnata*** L.

Characteristic compounds (Table 1)

Flavonoids-C-glycosides of apigenin (isovitexin) and luteolin (isoorientin)

In in vitro studies, passionflower extract showed affinity only for the GABA_A_ receptor. However, it did not affect the serotonin receptor and subreceptors [39]. Passionflower extract has also been included in clinical trials in patients with anxiety symptoms. Improvement comparable to that of oxazepam was demonstrated using objective methods, and the onset of action of the natural drug was also slower [40]. The anxiolytic effect of passionflower is well documented, but it is still uncertain what compounds in the extract are responsible for it. The derivative of pyron (maltol) from the herb inhibits spontaneous mobility in vivo, but in the extract, it is ineffective due to its low concentration and lability [41]. The anxiolytic effect of the alkaloids contained in the raw material has not been confirmed either. Probably the most effective are the C-glycosides of apigenin and luteolin [42]. Children 3–12 years of age can use passionflower only under medical supervision in a dose adjusted to the body weight [43].

***Piperis methystici rhizoma* (kava-kava)-*Piper methysticum* G. Forst**.

Characteristic compounds (Table 1)

Derivatives of pyron-kavalactones (methisticin)

In vitro studies have shown that the methysticin pepper rhizome affects the GABA_A_ receptor located in different regions of the brain: hippocampus, amygdala, medulla, frontal lobe, and cerebellum. It has been shown here that the effect of kavalactones is based on an increase in the number of receptor connections rather than changes in their affinity. It also turned out that the isolated kavalactones (including methisticin) have a weak affinity for the benzodiazepine receptor and GABA_A_ [44,45]. In human pharmacological studies, kava-kava has been shown to improve concentration. In the same study, the sedative-hypnotic effect was ruled out, even at a dose of 600 mg of methysticin pepper extract. These results were confirmed by the psychometric analysis, which showed an increase in concentration in the studied group of volunteers and the normalization of their emotional states [46]. In several clinical trials of kava-kava rhizome preparations, an improvement in patients with anxiety disorders, as measured by the Hamilton Scale (HAMA), of approximately 76% has been shown [47]. The clinical effect of the use of kava-kava in anxiety states, tensions, and nervous excitations, measured on various scales, showed a statistically significant improvement over the control groups [48]. Unexpected weakness, loss of appetite, weight loss, yellow discoloration of the conjunctiva of the eye and skin, dark urine, and colorless stools may signal liver damage. In such cases, treatment should be discontinued, and symptoms reported to a doctor. Combination with some benzodiazepines may cause a sign of extreme drowsiness. Few cases of liver side effects have been reported. Therefore, in order to minimize or exclude them, the drug should be used only in cases of moderate and moderately severe anxiety, tension, and nervous excitement states, not depressed states. The maximum daily dose of 120 mg of kavalactones should not be exceeded, and the duration of use should be one month or a maximum of two. Hepatic parameters (GPT and GT) should be checked before starting treatment and monitored once a week. The drug should do not used simultaneously with potentially hepatotoxic drugs, especially beta-blockers, antidepressants, and antimigraine preparations [49].


***Lupuli flos*-*Humulus lupulus* L.**


Characteristic compounds (Table 1)

Bitter acids-humulone

The essential oil–farnesene

Farnesene and lupulone are responsible for the healing effect on mood disorders such as restlessness, anxiety, and difficulty sleeping. 2-Methyl-3-buten-2-ol (a metabolic product of humulone, lupulone, and their derivatives) at high doses (800 mg/kg), given intraperitoneally to test animals, produced a CNS depressant effect by inducing deep sleep [50]. There are no data on the possibility of using the raw material during pregnancy and lactation. Only a doctor can decide to administer it. There are no objections to the use of hops flower with other drugs for the duration of use and no reports of adverse effects [51].


***Ballotae nigrae herba*–*Ballota nigra* L.**


Characteristic compounds (Table 1)

Phenylpropanoids in the glycoside and free-form acteoside

Flavonoids–derivatives of luteolin

The affinity of phenylpropanoids from the measles herb has been demonstrated in vitro to the benzodiazepine, dopamine, and morphine receptors. An in vivo study of the aqueous extract, administered parenterally, showed an anxiolytic and mobility-reducing effect [52]. The herb of measles may have a negative effect on driving a motor vehicle. Children aged 3 to 12 years under medical supervision can only take a proportion of adult dose according to body weight as nonalcoholic preparations [53].

### 2.2. Phytotherapy of Depression

Herbal drugs used in depression have an advantage over synthetic drugs because they maintain a balance between the levels of serotonin and norepinephrine in the central nervous system. Thanks to this, their use does not cause the risk of serotonin syndrome and bipolar disorder, as can occur with synthetic antidepressants that selectively raise the level of only serotonin or only norepinephrine in the central nervous system. The most important symptoms of classic depression, according to WHO and the International Classification of Diseases-10th Review (ICD-10), are depressed mood, feeling tired, unwillingness to perform even the simplest activities, and lack of interest and joy. With such symptoms, which can be described as apathy, CNS stimulants will be indicated.


***Hyperici herba*-*Hypericum perforatum* L.**


Characteristic compounds (Table 1)

Naftodiantrons-hypericin

Floroglucinols (a triphenol derivatives)–hyperforin

Biflavonoids-amentoflavone, biapigenin

Hydroalcoholic extracts of St. John’s wort herb in vitro weakly inhibit MAO-A and MAO-B but significantly the uptake of serotonin, norepinephrine, and dopamine. The stimulating effect of hyperforin on the NMDA receptor (N-methyl-D-aspartate), which is an activator of the receptor associated with the ion channel permeable to calcium, sodium, and potassium ions located in the synapses, acting as an agonist of the native (natural) ligand of the glutamate receptor has also been shown. The stimulation of the NMDA receptor produces a CNS stimulating effect that enables the combination of two types of stimuli: chemical (glutamic acid) and electrical (action potential) [54]. It is worth noting that the use of only high doses of the extract for at least 8 weeks increased the level of neurotransmitters in the hippocampus and decreased the metabolic turnover of serotonin here and in the hypothalamus [55]. Clinical trials comparing the effectiveness of St. John’s wort preparations with some synthetic antidepressants have shown that the efficacy of the plant material is not lower. Its therapeutic effect depends on the duration of the treatment and the daily dose for adults and children from 12 years: 450–1050 mg of dry extract daily. The optimal effect of using St. John’s wort herb occurs after 4–8 weeks (the beginning takes place after 8–14 days) [56]. Some in vitro studies indicate that naphthodiantrons, floroglucinols, and biflavonoids from herbs of St. John’s wort sensitize noradrenergic and serotonergic receptors, which may suggest such an antidepressant mechanism of this raw material. In vivo and in vitro studies indicate that naphthodiantrons and biflavonoids have effects on serotonergic, dopaminergic, and adrenergic receptors [57]. There is a possibility of the so-called serotonin syndrome when used simultaneously with other antidepressants. Contraindication to therapy with St. John’s wort herb is the simultaneous use of cyclosporine and other immunosuppressants and indinavir and other antiretrovirals. Patients should be monitored concomitantly using other antidepressants, anticoagulants, digoxin, and theophylline. St. John’s wort must not be used after organ transplants [56,58].

Herbal medicines have been long used to attenuate infectious diseases because of their lower side effects, low cost, and reduced potential to cause resistance. Many plant-derived essential oils can be effective in preventing and treating viral-induced respiratory tract infections. Salem and Ezzat [6] presented a review paper on applicable essential oils to prevent and treat COVID-19. This review showed further highlights that the antiviral properties of essential oils enable them to be engaged in the nutraceutical and pharmaceutical industries. The essential oils may be the future core for lead natural drug bio-discovery.

Depression is a heterogeneous mood disorder that has been classified and treated in various ways. However, many synthetic drugs are being used as standard treatment for clinically depressed patients. Authors Dhingra and Sharma believe that it is worthwhile to look for antidepressants from plants [59]. A number of medicinal plants have shown antidepressant properties by their therapeutic constituents. The causes of depression are decreased brain levels of monoamines such as noradrenaline, dopamine, and serotonin. Therefore, drugs restoring the reduced levels of these monoamines in the brain either by inhibiting monoamine oxidase or by inhibiting reuptake of these neurotransmitters might be fruitful in the treatment of depression. The present review focuses on medicinal plants and plant-based formulations having antidepressant activity in animal studies and humans.


***Rhodiolae radix*–*Rhodiola rosea* L.**


Characteristic compounds (Table 1)

Phenylpropane glycosides-rosavidin

Phenylethanoids–salidroside

In vivo studies of *Rhodiola rosea* roots showed an increase in the concentration of norepinephrine, serotonin, dopamine, and acetylcholine in the CNS in the tested animals, as well as the stimulation of ACTH secretion by the raw material extracts. Another study, based on the measurements of the concentration of biomarkers signaling symptoms of fatigue (including lactate dehydrogenase, fatty acid synthesize, and temperature shock proteins), showed that *Rhodiola rosea* extract significantly increased liver glycogen concentration, temperature shock protein expression, and the level of oxygen compounds, and decreased the concentration of aspartate aminotransferase. This indicates that the organism of the tested animals increased its efficiency during, for example, a swimming test and decreased the level of these enzymes, which would indicate the state of exhaustion during exercise [60]. Pharmacological studies of Rhodiola root in humans showed a statistically significant improvement in cognitive disorders and a decline in physical performance. The properties of this raw material in counteracting fatigue have been demonstrated in subsequent trials, in which the effects of strengthening muscles, as well as improved reaction speed, better concentration, greater mental performance in solving mathematical tasks, and more excellent mental resistance, compared with the placebo group, were observed [61]. Most of the results were obtained by objectively measured methods. It is suggested that the raw material can be used in the treatment of dementia, psycho-physical exhaustion, and as an aid in unipolar depression [62].

### 2.3. Herbal Medicines in the Mixed Anxiety–Depressive Syndrome

Anxiety and depression problems are among the most common mental disorders. According to the International Statistical Classification of Diseases and Related Health Problems (ICD-10) and the Diagnostic and Statistical Manual of Mental Disorders (DSM-IV), both conditions are treated separately. However, there is an opinion among clinicians that anxiety and depression are common categories of mood disorders. According to one meta-analysis of patients with generalized anxiety disorder, one-third had symptoms severe enough to be diagnosed with a depressive episode. These data have been confirmed in other studies, which justifies the increasingly frequent suggestion among doctors that the coexistence of anxiety–depressive syndromes is the rule rather than the exception and is still the subject of research and analysis [63]. The diagnostic criteria-based classification system (DSM-IV-TR) identifies anxiety–depressive disorders as the dysphoric mood that is persistent (or recurrent) for at least one month with at least four of the following symptoms: difficulty concentrating or feeling light-headed, sleep disturbances (difficulty falling asleep, interrupted sleep, restless sleep), fatigue or lack of energy, irritability, worrying, excessive vigilance, predicting the worst, feeling of lack of perspective, and poor self-esteem [64]. Based on the pharmacological properties of plant materials acting on the central nervous system, the following may be helpful in depression–anxiety syndromes: valerian root (especially when anxiety episodes are prevalent), lavender flower and oil, plant materials containing caffeine, and aromatherapy with Melissa oil.


**
*Lavandulae flos*
**



***Lavandulae aetheroleum-Lavandula officinalis* Chaix**


Characteristic compounds (Table 1)

Essential oil-linalool acetate, linalool

Lavender oil and its two main compounds inhibited, with dose-dependent intensity, electrically induced isolated nerve cells in animals. In in vivo studies, lavender oil was administered parenterally, and by inhalation, dose-dependently inhibited the convulsions induced in animals by 60–70%. In another study, lavender oil parenterally administered to animals reduced their spontaneous mobility and excitability induced by caffeine benzoate. A similar effect was obtained after 60 min of inhalation of lavender oil, linalool acetate, and linalool by 78%, 69%, and 73%, respectively, compared with the control group. An anxiolytic effect of lavender oil was also observed after oral administration in test animals [65]. A pharmacological study involving humans administered three drops of 10% lavender oil (in grape seed oil) on a cotton roll for 3-min inhalation at a distance of 8 cm from the nose showed a reduction in beta waves in the EEG, which may be indicative of drowsiness caused by the given substance. In the same study, a statistically significant improvement in mood (observation of the Profile of Mood States) of 56% and a reduction in anxiety states (observation of the State-Trait Anxiety) of 9% were observed in the subjects. This study’s detailed analysis of the EEG recordings showed significant changes in the chart, showing more of an extroverted effect of lavender oil than a depressive effect on the CNS [66]. A statistically significant improvement was also demonstrated in healthy volunteers subjected to memory tests and reaction time trials compared with the control group. These results were confirmed in a clinical study involving patients with depression who were treated with lavender flower tincture. There was a significant improvement as measured by the Hamilton scale (from 19 to 12 points), comparable to that of imipramine (from 19 to 9 points) [67]. Based on pharmacological and clinical studies, it can be concluded that lavender oil and lavender flower work to improve mood, have an antianxiety effect, and are more antidepressants than a sedative. The lack of research excludes the use of lavender flowers and lavender oil during pregnancy and lactation and in children under 12 years of age [68].


***Paulliniae semen* (guarana)–*Paullinia cupana* Kunth ex H.B.K. var. *sorbilis* (Mart.) Ducke (*P. sorbilis* C. Mart.)**


Characteristic compounds (Table 1)

Methylxanthines-caffeine (1.1–5.8%)

Aqueous extracts of guarana seeds and their xanthine fraction inhibited platelet aggregation in a dose-dependent manner by approximately 70% in comparison with the control material. This effect was confirmed in an in vivo study after parenteral administration of 10% water extract of guarana seeds. Another study involving animals injected with a liver-damaging agent showed the cytoprotective effect of powdered seeds [69]. Further in vivo studies of guarana showed its stimulating effect. In the swimming test, this material significantly shortened the time of immobility and increased the activity of the tested animals, which proves its antidepressant-like effect. It was found that the main active compound of the raw material, at a dose of 20 mg/kg body weight, blocks the action of the adenosine antagonist, a compound responsible, inter alia, for energy transport. Paulini seed extract also inhibited scopolamine amnesia and reduced obesity in vivo [70]. In pharmacological studies involving people aged 20–35 years, powdered guarana seeds improved cognitive processes and decreased anxiety and sleep disorders. Other studies showed, by means of objective methods, the improvement of mood, concentration, and increased reaction speed in the studied volunteers. It has been shown in human studies that guarana increases self-confidence [71]. The daily dose must not exceed 3 g of guarana preparations. Hypertension and age up to 16 years of age are a contraindication to the use of the raw material. Patients with stomach ulcers may experience discomfort after using Paulini seeds. Because of the caffeine content, caution should be exercised in glaucoma, combination with psychoanaleptics and alcohol. There may be vomiting, diarrhea, or heart palpitations after using the preparation with 15 g of raw material. Raw material can cause sleep disturbance and overstimulation [72].

### 2.4. Medicinal Plants That Improve Memory Processes and Cognitive Functions

Changes in the immune system have been seen in studies of the cerebrospinal fluid, which bathes the brain, in people who have been infected by SARS-CoV-2. This includes the presence of antibodies—proteins made by the immune system to fight the virus—that may also react with the nervous system [73]. This can cause disturbances in the functioning of neurons in the central nervous system as protein, and amyloid-forming proteins accumulate, thus manifesting symptoms of Alzheimer’s disease. Problems in communicating with other people, forgetting topics of conversation, inability to repeat conversations, and inappropriate interruptions during a conversation, are the most important symptoms of a diagnosis of early dementia. Characteristic is the inability to find things at home, repeating remarks and questions many times, losing the thread of conversation, and repeatedly performing the same activities. The early stages of dementia are the inability to learn new things, short-term memory impairment, difficulties in “finding yourself” in new places, and impairment of cognitive processes, such as concentration, perception, thinking, and planning [74].

These are the reasons for the deterioration in well-being, mood swings, and difficulties in carrying out daily activities. Among the plant materials used as an auxiliary in memory disorders and cognitive impairment are those that have a similar mechanism of action to synthetic drugs used in Alzheimer’s disease (increase the concentration of acetylcholine in the central nervous system) and also protect nerve cells, accelerate time reactions, increase interest in the environment, and improve mood.


***Ginkgo folium*-*Ginkgo biloba* L.**


Characteristic compounds (Table 1)

Trilactone diterpenes-ginkgolides

Sesquiterpene lactones-bilobalide

Ginkgo leaf has a two-step (dose-dependent) effect on labeled serotonin. Doses of 4–16 g/mL cause a significant increase in reuptake of this compound, while doses higher than 32 g/mL reduce this process. In other studies, induced by various factors, ginkgo leaves have been shown to protect the processes of serotonin and dopamine reuptake inhibition in cerebral cortex cells, and bilobalide significantly increases choline release. In vitro studies have demonstrated the protective potential of the extract of cultured hippocampal nerve cells against the toxic effects of amyloid plaques (pathological protein deposited in CNS nerve cells) and against apoptosis. Among other things, we demonstrated an inhibitory effect on the formation of senile toxic tau protein (in the form of neurofibrillary tangles) in the hippocampus, counteracting the induced cerebral hypoxia in test animals and counteracting artificially induced hemiplegia [75,76]. In human pharmacological studies, the EEG graph showed a significant increase in alpha wave activity and an improvement in the type of cognitive response. In a psychometric test, improvements in functional memory were demonstrated after volunteers were given 50 to 240 mg of ginkgo leaf extract. These results were confirmed in another study with the same dose and form of the raw material—the effect measured by the EEG graph was similar to that of tacrine (a drug formerly used in Alzheimer’s disease)—and a percentage increase in alpha waves was observed. In several studies with the participation of volunteers, a positive effect of the plant drug on executive memory processes was demonstrated—improvement in the speed of data reproduction and better data consolidation [77]. In studies on the effectiveness of the effect on cognitive processes, it was shown by objective methods, measuring the speed of reaction, the precision of remembering and reproducing, increased speed of performing memorized actions, and consolidation of executive memory [78,79]. In clinical studies of ginkgo leaf extract with patients with moderately advanced dementia, a 25% improvement in the psychometric Crichton scale test was obtained after 3 months of treatment. Similar results were obtained in more than a dozen other studies in dementia patients aged 50–75 years. The various scales showed an improvement under the influence of ginkgo extract of about 25% [80].


***Lavandulae aetheroleum*–*Lavandulae flos* Chaix**


Characteristic compounds (Table 1)

Essential oil-linalool acetate, linalool

In vivo studies showed that lavender oil improved memory function in scopolamine-treated animals [81]. In a pharmacological study involving 144 volunteers, the effects of lavender oil on cognitive function were evaluated using computerized methods. A statistically significant improvement in functional memory, reaction speed, and concentration was demonstrated. In another study, inhalation of lavender oil caused a reduction in anxiety and an increase in the state of arousal, as assessed by a special scale. The latter effect was confirmed in a subsequent study, in which lavender oil administered to volunteers in the form of a spray (0.29 mg/L) was shown to increase concentration time and enable them to focus attention significantly longer compared with the control group. Similar results were obtained on various scales assessing attention, concentration, and reaction time [82,83].


***Melissae aetheroleum*–*Melissae flos* L.**


Characteristic compounds (Table 1)

Monoterpene aldehydes–citronellal

In a clinical trial involving 72 patients, with a mean age of 78 years, with clinically diagnosed excessive agitation due to severe dementia, Melissa oil was administered as a 10% lotion lubricated on the neck and both arms. On the Cohen–Mansfield Scale, which assesses changes in agitation, there was an average of 35% improvement (versus 11% improvement in the placebo group). Moreover, objectively assessed improvements in quality of life showed a statistically significant difference in the verum group versus the placebo group [84]. In pharmacological studies with humans, an alcoholic extract of lemon balm leaves, administered as a 10% cream, applied in the neck area in a single dose for 7 days with a toilet break, showed improvement in concentration and improvement in memory processes in objective and subjective evaluations [85,86].


***Murrayae folium*–*Murraya koenigii* L.**


Characteristic compound (Table 1)

Carbazole alkaloid–mihanimbine

Mahanimbine, isolated from the leaves, has been shown to be an inhibitor of acetylcholinesterase, an enzyme that breaks down acetylcholine. Decreased concentration of this neurohormone in the CNS is responsible for memory impairment—the first symptom of dementia. Another study tested the properties of murai leaf extract on cognitive function, serum cholesterol levels, and cholinesterase activity in the brain. Administration of the extract for 30 days showed improved memory in vivo on an exteroceptive and interoceptive model. Scopolamine- and diazepam-induced amnesia was found to be reduced with dose-dependent intensity. A reduction in cholinesterase activity and total cholesterol concentration was also observed [87].

## 3. Phototherapy of Cardiovascular and Heart Disorders as a Result of Neurological Symptoms in COVID-19 Infection

Cardiac injury in patients infected with the novel Coronavirus (COVID-19) seems to be associated with higher morbimortality. In addition to disorders that COVID-19 causes to the cardiovascular system, more protection should be employed for patients with preexisting cardiovascular disease. Hence, patients with COVID-19 must be rapidly treated to reduce mortality [88]. Emotional states have a major impact on cardiovascular function. The stress of everyday life puts a strain on the cardiovascular system. The consequences of this can appear unnoticed. Disregarding the first symptoms of cardiovascular problems, such as shortness of breath, fatigue disproportionate to the effort, and shortness of breath may necessitate using preparations based on cardiac glycosides—strong-acting drugs with a low therapeutic index. This is associated with the risk of adverse effects, and neglect of treatment may also lead to irreversible changes in the heart and coronary vessels. Plant preparations are important in the prevention and auxiliary treatment of cardiovascular diseases [89].


***Crataegi fructus*-*Crataegus monogyna* Jacq. and *C. oxyacantha* L.**


Characteristic compounds (Table 1)

Oligomeric proanthocyanidins with epicatechin as the major component

Anthocyanins contained in the fruits of hawthorn have an affinity for the heart muscle. In vitro studies have shown that the likely mechanism of action of hawthorn raw materials involves inhibition of c-AMP phosphodiesterase activity, thromboxane biosynthesis, stimulation of prostacyclin biosynthesis, and reduction in cell membrane permeability to Na+, K+, and ATP-aze enzyme [90]. The cardiotonic effect of hawthorn fruit has been confirmed by many studies in vitro, pharmacological studies with animals, and clinical trials. On their basis, it can be concluded that preparations from hawthorn strengthen the force of cardiac muscle contraction (inotropic effect +); cardiac glycosides have a similar effect, but unlike them, hawthorn raw materials have a more favorable ratio between the contraction frequency and its force. Their action results from direct stimulation of receptors, as well as from improvement of energy metabolism in the heart muscle. They affect heart rate, primarily by prolonging the refraction time (time of insensitivity to action potential and stimulus to heart contraction) and have antiarrhythmic effects (in vivo studies) [80]. Hawthorn anthocyanins increase the amplitude and time of myocardial contractility. This was demonstrated in an ex vivo study on an isolated heart and confirmed by administering to adult patients the extract of leaves and flowers in the concentration of 90–180 g/mL, which resulted in significant prolongation of refraction time (it is worth mentioning that cardiac glycosides shorten refraction time by acting directly on Purkinje fibers in the myocardium and indirectly, by activation of the sympathetic system and thus increasing noradrenaline concentration) [91]. The aforementioned properties found in vitro and in vivo allow the use of hawthorn preparations in mild bradyarrhythmia and paroxysmal tachycardia. Here, hawthorn preparations have rather unique properties. Usually, compounds with inotropic positive activity are arrhythmogenic; however, hawthorn anthocyanins show properties that stabilize heart rhythm [92]. Hawthorn preparations may be used only under the supervision of a cardiologist in heart failure corresponding to the I and II stage, according to NYHA (New York Heart Association). The efficacy of hawthorn preparations in patients with the abovementioned symptoms has been confirmed in clinical trials. There was a statistically significant effect of better tolerance to physical effort, reduction in palpitations, dyspnea on exertion, an increase in ejection fraction by 6.7%, and a slight decrease in blood pressure. The efficacy of hawthorn preparations in stage II symptoms indicating congestive heart failure is noteworthy [93].

Objective and subjective methods noted the improvement. Hawthorn preparations are also recommended to prevent the risk of circulatory insufficiency and treat chronic conditions of the so-called senile heart [94].

## 4. Herbal Raw Materials That Inhibit Inflammation of the Respiratory Tract Caused by COVID-19 Infection

COVID-19 is an infectious disease that can affect people of all ages in many ways. It is most dangerous when the virus spreads from the upper respiratory tract into the lungs to cause viral pneumonia and lung damage leading to Acute Respiratory Distress Syndrome (ARDS). When severe, this impairs the body’s ability to maintain critical oxygen levels in the bloodstream—which can cause multiple body systems to fail and can be fatal [95]. The herbal resources discussed below may also be used for the prophylaxis of respiratory tract inflammation in individuals who are concerned about such an adverse reaction after COVID-19 vaccines.


***Hederae helicis folium*-*Hedera helix* L.**


Characteristic compounds (Table 1)

Triterpene saponosides-hederacoside C, α-hederine

Hederine, formed from hederacoside C, indirectly stimulates adrenergic receptors present in smooth muscle from the trachea to terminal vesicles for the production (in the form of a complex of dipalmitolecithin and a carrier protein—apoprotein) of surfactant, a natural surfactant composed of glycerophospholipids, inside the alveoli. There is a relaxation of smooth muscles, a reduction in the viscosity of airway secretions, an improvement in ciliary-mucosal clearance, and a reduction in elastic resistance to lung respiratory work [96]. Several clinical trials involving large numbers of patients of different age ranges have found ivy leaf extracts to be effective in chronic airway obstruction, chronic bronchitis, pulmonary lesions, and acute bacterial respiratory infections. No data are available for use during pregnancy and lactation. There is a real danger of acute contact dermatitis if in contact with the plant. Use caution in patients with gastric and duodenal ulcers. Because of the lack of data, use in children under 4 years of age is not recommended [97].


***Polygalae radix*-*Polygala senega* L.**


Characteristic compounds (Table 1)

Triterpene saponosides-presegenin glycosides

In vitro studies showed inhibition of cyclooxygenases 1 and 2 by the raw material extract (at a concentration of 12.5 L/mL) and inhibition of the *Herpes*-type virus and influenza A2 virus. Intragastrically administered cruciferous root extract induced increased respiratory secretion in the test animals. This study confirmed, through the gastric mucosa, the indirect mechanism of action of this saponoside raw material [98]. Pharmacological studies in humans of the liquid extract of the roots of cruciferous virginia have shown secretolytic and mucolytic effects suggesting its efficacy in facilitating expectoration [99].


***Pelargonii radix*-*Pelargonium sidoides* DC.**


Characteristic compounds (Table 1)

Derivatives of coumarin compound umckalin

Sulfur-oxygen derivatives-6,7,8-trihydroxybenzopyranone disulfate

Tannins composed of proanthocyanidins

African geranium root extract exhibited potent antimicrobial activity—antibacterial and antiviral. The plant preparation interfered with the penetration of these microorganisms into the host cell [100]. In a clinical study, patients with radiographically proven acute inflammation of the nasal sinuses and nasal mucosa caused by the bacterial infection were treated with ethanolic (11%) African Pelargonium root extract (1:8–10) at a dose of 60 drops for a maximum of 22 days. A statistically significant difference was obtained between the groups using the plant drug versus placebo, with good tolerance of the extract. The antibacterial properties of African geranium root may also be due to increased phagocytosis in peripheral blood [101,102]. Preparations of African geranium roots are contraindicated in people with blood clotting problems. There are no data on the possibility of using the raw material in pregnant and breastfeeding women, in children under 12 years of age, and in people with severe problems with parenchymal organs [103].


***Lichen islandicus*-*Cetraria islandica*(L.) Acharius**


Characteristic compounds (Table 1)

Polysaccharides hot-water soluble D-glucans and cold-water soluble D-glucans, as well as branched combinations of D-glucose and glucuronic acid

Lichenic acids, fumaroprotocetric acid, and protolichesteric acid

Anti-inflammatory effects have been demonstrated by both the extract of Icelandic Sclerotium (obtained by hot water extraction) and lichenic acids. In vitro, they inhibited leukotriene B4 biosynthesis by inactivating 5-lipoxygenase. Protolichesteric acid showed particular activity [104]. A clinical study of an aqueous extract of Iceland lichen husks, in a dose equivalent to 0.5 g of raw material, showed a statistically significant therapeutic effect in a group of test patients with inflammation and dryness of the airway mucous membranes, markedly enlarged lymph nodes, a coated tongue, and hoarseness [105]. In another study involving 100 patients, lozenges containing 160 mg of Iceland lichen positively affected 86 acute and chronic inflammation cases of the bronchial and pharyngeal mucous membranes [106].


***Plantaginis lanceolatae folium et herba*–*Plantago lanceolata* L.**


Characteristic compounds (Table 1)

Iridoid glycosides-aucubin

Phenylethanoids–acteoside

In vitro alcoholic extracts of plantain herb showed anti-inflammatory effects. Phenylethanoids appeared to be responsible for this effect and were attributed to the inhibition of 5-lipoxygenase, one of the enzymes that catalyze the transformation of arachidonic acid. It was proven that acteoside revealed an anti-inflammatory effect by inhibiting the biosynthesis of 5-hydroxyeicosatetraenoic acid and leukotriene B4 (inflammatory factors) in isolated human leukocyte cells. The preparations obtained by maceration of the raw materials with cold water showed inhibitory effects on the growth of dangerous pathogenic bacteria. The anti-inflammatory activity of both raw materials and aucubin isolated from them was confirmed by in vivo studies [107].

All discussed herbal medicines and their dominant compounds are shown in Table 1.

**Table 1 cells-11-01897-t001:** Chemical structure of main chemical compounds in medicinal herbs for the relief of neurological, cardiovascular, and respiratory symptoms after COVID-19 infections.

Medicinal Herbs	Active Molecule	Structure	Mechanism
** *Valerianae radix* **	Valeranone	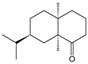	Extract significantly decreased ^3^H-glutamate binding [108]
Valtrate	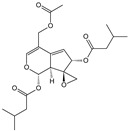	Extract influences NMDA and AMPA receptor binding [108]
** *Melissae folium* **	Apigenin	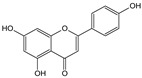	Interaction with both S 1 and S2 domains of the spike protein of SARS-CoV-2Binding affinity with the SARS-CoV-2 major protease (6 LU7) [109]
** *Melissae flos* **	Citronellal	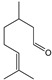	Inhibition of ACE2 activity [110]
** *Passiflorae herba* **	Isovitexin	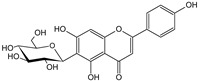	SARS-CoV-2 main protease inhibitor [111]
Isoorientin	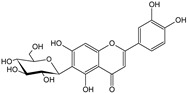	
** *Piperis methystici rhizoma (kava-kava)* **	Methysticin	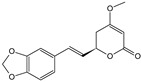	SARS-CoV-2 main protease inhibitor [112]
** *Lupuli flos* **	Humulone	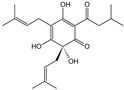	Act as immunomodulator [113]
Farnesene	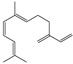	
** *Ballotae nigrae herba* **	Acteoside	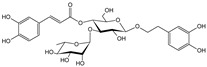	SARS-CoV-2 main protease inhibitor [114]
** *Hyperici herba* **	Hypericin	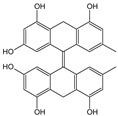	Block of the expression of the SARS-CoV-2 nucleocapsid protein (N) [115]
Hyperforin	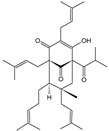	
Biapigenin	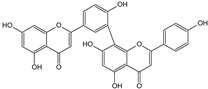	Block of the expression of the SARS-CoV-2 nucleocapsid protein (N) [115]
** *Rhodiolae radix* **	Rosavidin	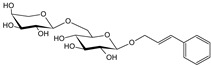	Transcriptional control of metabolic regulation [116]
Salidroside	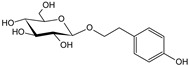
** *Lavandulae flos* **	Linalool acetate	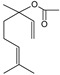	Inhibition of ACE2 activity [110]
Linalool	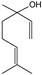	
** *Pauliniae semen* ** **(guarana)**	Caffeine	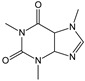	Inhibition of ACE2 activity [117]
** *Ginkgo folium* **	Ginkgolide	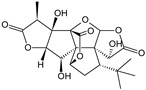	
Bilobalide	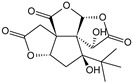	Inhibition of the targeting protein and DNA [118]
** *Murrayae folium* **	Mahanimbine	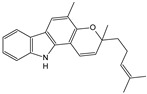	Inhibition BACE1 and AChE,Decrease IL-1 β and TNF-α, COX2,Increase TGF-β and IL-10 [119]
** *Crataegi fructus* **	Epicatechin	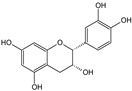	Inhibition of ACE2 activity [120]
** *Hederae helicis folium* **	Hederacoside C	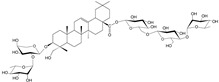	Inhibition of ACE2 activity [121]
α- hederine	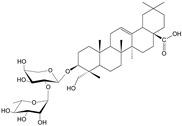
** *Polygalae radix* **	Presegenin	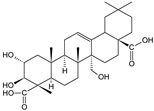	An inhibitor of both S and ACE2 proteins [122]
** *Pelargonii radix* **	Umckalin	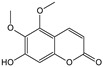	
6,8-Bis(sulfooxy)-7-methoxy-2H-1-benzopyran-2-one	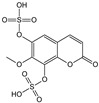	Affects propagation [123]
** *Lichen islandicus* **	Fumaroprotocetric acid	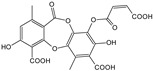	Stimulate an increase of NO release in macrophages [124]
Protolichesteric acid	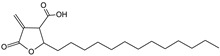	
** *Plantaginis lanceolatae folium et herba* **	Aucubin	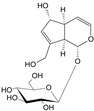	Inhibition of TNF-alpha and IL-6 production [125]
Acteoside	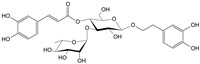

The COVID-19 pandemic is showing more and more risks. A year and a half after the first cases were reported, symptoms of dementia (irreversible) began to be observed, even in people with a mild course of infection. This is the result of a decrease in the level of acetylcholine, responsible for memory processes. Alzheimer’s disease is irreversible, and it is very important to start using early drugs that prevent the breakdown of acetylcholine by inhibiting acetylcholinesterase. Herbal medicines can be a life saver. The most important herbal plants for the treatment of neurological, cardiovascular, and respiratory symptoms after COVID-19 have been discussed in the paper. Some of them are shown in Figure 2.

Ginkgo biloba leaf, in particular, has a very good pharmacological profile (Figure 2A).

It seems essential to use this drug even prophylactically and not wait for the first signs of dementia in a diagnosed COVID-19 infection [118]. Anxiety and depression are equally dangerous ailments associated with COVID-19 virus infection, and it is imperative to treat them. Among plant medicines, the first choice for anxiety is *Valeriana officinalis* root (Figure 2B).

In addition to the advantages of this drug and the advantage over synthetic benzodiazepines mentioned in the introduction, it should be emphasized that when using valerian root, no tachyphylaxis is observed [34]. However, in the case of depression, the first choice among all drugs, including synthetic ones, should be *Hypericum perforatum* herb (Figure 2C).

Its mechanism of action of increasing both serotonin and norepinephrine levels provides the effectiveness of synthetic drugs while not causing the dangers they can cause [58]. There are cases of CNS disorders where the primary symptoms of anxiety and depression are present. Some of their symptoms are common (insomnia). In such cases, neurologists and psychiatrists may turn to lavender flowers (Figure 2D) or aromatherapy with lavender oil when symptoms of depression predominate over anxiety or valerian root when anxiety predominates over depression [126].

It is known that disorders of the central nervous system affect the heart and circulatory system with life-threatening force. Cardiologists have at their disposal only drugs of plant origin (there is a lack of synthetics on the pharmaceutical market). Therefore, in the case of these disorders, the first choice is hawthorn fruit (Figure 2E) [94].

This raw material contains anthocyanins, which have a beneficial mechanism of action in depression. The effect of that compound consists in stimulating the receptors located in the heart. Thanks to this, while having a positive inotropic effect (the force of heart muscle contraction is increased), and the refraction time is prolonged. Therefore, under the influence of hawthorn fruits, the heart works more efficiently and has a longer time for “rest” [127]. Major threats to the health and life of COVID-19 patients are respiratory system disorders. Here, plant medicines should be used rather prophylactically and at the sign of first symptoms of respiratory disorders. Ivy leaf (Figure 2F) in such cases seems to be a very good choice.

Its natural compounds primarily dilate the smooth muscles of the respiratory tract and dilute the mucus lingering there, provoking irritation of the gastric mucosa and its effective removal [128]. Continuous mutations of COVID-19 force the search not only for drugs that inhibit the symptoms of the infection but also for agents that act against coronaviruses. Plant-based drugs, too, may have an important part to play in the fight against the COVID-19 pandemic [129].

Chojnacka et al. [130] showed that in the recent period, research has focused on investigating which components of polyphenolic extracts inhibit viruses of various origins and determine the inhibitory doses. Polyphenols affect viruses through the viral replication cycle, gene expression reduction, viral structure changes, interaction via RNA-dependent RNA polymerase, etc. These interactions depend on the origin of the virus, the type of polyphenolic compounds used, and their synergistic interaction. The synergistic interaction of two or more polyphenols as a drug can improve the inhibitory effect, even if they have little or no antiviral activity separately. Authors suggest that the unambiguous identification of the inhibitory mechanism of these compounds is the main issue for which comprehensive studies should be carried out in order to determine the relationship between given groups of substances and their effects on viruses. One of the promising plants could be *Stizolophus balsamita* (Lam.) K. Koch. rich in flavonoids [131].

Plants have been used to treat viral infections for a long. Literature review affirms that herbal medicines show scientific evidence for host protection against various types of coronaviruses, with a focus on SARS-CoV-2. Some of them stabilize the Renin-Angiotensin-Aldosterone System, which is associated with the entry of the SARS-CoV-2 into human cells; however, it would be unwarranted to declare that all these plants can be blindly prescribed in patients with viral infections [132,133]. Therefore, it is required to establish and validate all the assessment parameters such as herb–drug and herb–herb interaction and toxic effects following the pharmacological authentication of plant materials for human consumption. Because of the side effects of synthetic medicine, the world has started looking toward herbal medicine for the treatment of viral diseases, which are comparatively more accessible and economical and bear fewer chances of toxicity and resistance.

## 5. Conclusions and Future Perspectives

Phytopharmaceuticals, herbal or herbal drugs, are pharmaceutical preparations with proven therapeutic effects; according to the European Scientific Association for Phytotherapy, they are products useful in medicine, active ingredients of which are medicinal plants, their parts or substances derived from them, or combinations mentioned in a processed form. Dietary supplements containing ingredients from medicinal plants are classified as food products; their task is to supplement the daily diet with components supporting the physiological processes taking place in the body. Biopharmaceuticals are produced by living cells and are therefore widely accepted by the ecological society. Plant biopharmaceuticals can be produced by in vitro cultivation, as well as by plants growing on farmland (molecular pharming) [134,135,136]. Plant substances can be very helpful in inhibiting the dangerous symptoms of COVID-19 infection. Cardiovascular and respiratory plant resources protect the nervous system from the symptoms of COVID-19 infection.

Because of multiple active phytochemicals acting synergistically on various stages of viral replication, medicinal plants appear as a hope for COVID-19 complications treatment. However, only in silico studies with computer-aided docking and a few in vitro models are available, which show their anti-SARC-CoV-2 activities. Thus, in the future, more in vivo studies, especially clinical trials, are needed to establish the safe dose and clinical efficacy of their active compounds. Interestingly, a chemo–herbal combination in the COVID-19 therapy is suggested. In this approach, medicinal plants are administered with synthetic drugs, which could be a more feasible and effective approach to combat this viral pandemic.

The selection of medicinal plants, the development of methods for obtaining bioactive substances from them, or the preparation of a standardized extract is the first step in the process of obtaining biodrugs from plants. The next step, apart from clinical trials, on this path is the research on the way of delivering biodrugs to the organism. In this area, significant changes related to nanotechnology and control methods of drug delivery have been observed in recent years.

Medicinal plants and active chemical compounds presented in this study seem extremely useful in long COVID, where numerous neurological complications appear, e.g., COVID-related sleep disturbance.

## Figures and Tables

**Figure 1 cells-11-01897-f001:**
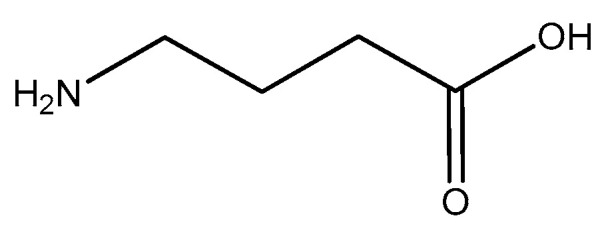
Gamma-aminobutyric acid (GABA).

**Figure 2 cells-11-01897-f002:**
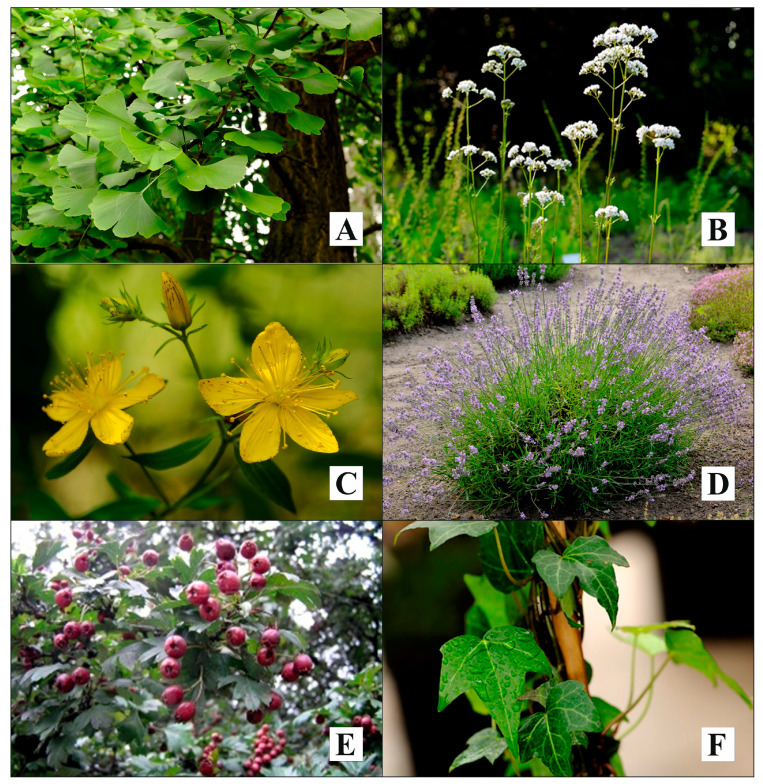
Chosen plants in the treatment of COVID-19 complaints (Garden of Medicinal Plants in Department of Medicinal and Cosmetic Natural Products, Poznan University of Medicinal Sciences, 33 Mazowiecka Street, 60-623 Poznan, Poland): *Ginkgo biloba* (**A**), *Valeriana officinalis* (**B**), *Hypericum perforatum* (**C**), *Lavandula officinalis* (**D**), *Crataegus monogyna* (**E**), *Hedera helix* (**F**).

## Data Availability

This study did not report any data.

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
