# Peer review of "Medicinal Herbs in the Relief of Neurological, Cardiovascular, and Respiratory Symptoms after COVID-19 Infection A Literature Review"

_cells, 2022, doi:10.3390/cells11121897_

Round 1

Reviewer 1 Report

Nawrot et al attempted to review medical herbs in the relief of neurological, cardiovascular and respiratory symptoms after COVID-19 infection.  They described potential complications of COVID-19 disease in humans and introduced medical herbs in treatment of neurological, cardiovascular and respiratory diseases, followed by segmenting the potential therapeutic effects of many medical herbs on the major complications of COVID-19 disease in these systems. Although this topic is interesting, however I have following concerns regarding the manuscript.

Basically, currently, there is no systemic study on how many people would be affected by COVID-19 related complications in the literature; there is no information on the potential mechanisms by which COVID-19 causes the complications in the literature; there is also no standard clinical trial to test the therapeutic efficacy and safety of any medical herb  in treatment of COVID-19 related complications in the literature. Accordingly, all discussion in the manuscript was based on previous publications, which may mislead clinicians due to potential difference in the pathogenic process and mechanisms of these complications with traditional diseases. Thus, a medical hypothesis will be more suitable.

Conceptually, this manuscript did not bring much new scientific information on our scientific community and many information is already reviewed by publications in the literature. There is no information on how these medical herbs relieve clinical symptoms in the manuscript. There is no author’s perspectives, current challenges and future studies in the manuscript. There are also some mistakes. For example, benzodiazepines are GABAA receptor PAM, but not agonists and they can bind to TSPO to regulate many biological functions.

Structurally, the introduction appeared to be too long and should separated into two sections, brief introduction of the rationale and aims of this review manuscript and the second one of COVID-19 related complications in humans to clarify the novelty and potential significance of this study.

Finally, there are some unnecessary repeated sentences and repeated definition of aberrations. Terminologically, there are wordings and sentences that are not professional. I do not feel these pictures are necessary. If included, these pictures may be combined. It is important to update the references.

Reviewer 2 Report

Current review article described the medicinal plants in the relief of neurological, cardiovascular, and respiratory symptoms after COVID-19 infection. Please conduct the concerns below.

  1. The plant materials and their dominant compounds were not indicated in the abstract.
  2. In the introduction, one useful product named NRICM101 which is a traditional Chinese medicine formula was not included. Please find it from Biomedicine & Pharmacotherapy 133 (2021) 111037 by Tsai K.C. et al. to introduce in clear.
  3. Please show one table to list all dominant compounds described in current report.
  4. Changes of each compound after COVID-19 infection were not conducted. Why?
  5. Conclusions were not indicated. What is the useful plant for COVID-19 infection?
  6. Please revise the references using the scientific journals but not the specific reports that were hard to find.

Reviewer 3 Report

Dear Sir

I just went through manuscript.It is well written.However authors.I recommend its publication in current form

Reviewer 4 Report

The Cells manuscript cells-1721923 “Medicinal herbs in the relief of neurological, cardiovascular, and respiratory symptoms after COVID-19 infection. A Literature Review” by Joanna Nawrot review the potential beneficial effect of phytotherapy for the treatment of neurological symptoms in COVID-19 infection.

The manuscript is well written, the literature was reviewed deeply.

Information are reported clearly and well discussed.

A careful editing of the proof is required since the dimension of the character typo are inconsistent.

The placement of the figures should be edited.

Line 91: and HBV (Hepatitis B virus): it seems that there is a different dimension of the text

Line 172: character dimension

Line 250: character dimension

Round 2

Reviewer 2 Report

It is easier to read in this new form.